# Epidemiology, Risk Factors for Gastric Cancer and Surveillance of Premalignant Gastric Lesions: A Prospective Cohort Study of Central Saudi Arabia

Georgios Zacharakis [1,*] , Abdulaziz Almasoud [2,3], Omar Arahmane [4], Jamaan Alzahrani [5] and Sameer Al-Ghamdi [5]

1 Endoscopy Unit, Department of Internal Medicine, College of Medicine, Prince Sattam Bin, Abdulaziz University, PrinceSattam Bin Abdulaziz University Hospital, Al-Kharj 16278, Saudi Arabia
2 Department of Gastroenterology, Prince Sultan Military Medical City, Riyadh 11159, Saudi Arabia; aoalmasoud@psmmc.med.sa
3 Endoscopy Unit, Al-Kharj Military Hospital, Al-Kharj 11494, Saudi Arabia
4 Endoscopy Unit, King Khaled Hospital and Prince Sultan Centre for Health Care, Al-Kharj 11942, Saudi Arabia; arahmane_o@hotmail.com
5 Department of Family and Community Medicine, College of Medicine, Prince Sattam Bin Abdulaziz University, Al-Kharj 16278, Saudi Arabia; j.alzahrani@psau.edu.sa (J.A.); sh.alghamdi@psau.edu.sa (S.A.-G.)
* Correspondence: g.zacharakis@psau.edu.sa

**Abstract:** (1) Background: Saudi Arabia (SA) is a country with a low incidence of gastric cancer (GC). In this study, we sought to assess the epidemiology of GC, its clinicopathological profiles, and its association with risk factors as well as to identify premalignant gastric lesions (PGL) and examine neoplastic progression. (2) Methods: This five-year prospective study screened for GC and PGL in asymptomatic Saudi patients, aged 45–75 years (n = 35,640) and living in Al Kharj, Riyadh province in central SA. Those who were positive in a high-sensitivity guaiac fecal occult blood test (HSgFOBT+) and had negative results in colonoscopy offered to undergo upper GI endoscopy (n = 1242). Factors associated with GC were examined. (3) Results: The five-year participation rate was 87% (1080/1242). The incidence rate of GC was 26.9 new cases per 100,000 population per year (9.6 new cases per year/total population at risk—35,640), and it was 8.9 cases per 1000 persons per year among the 1080 subjects with HSgFOBT+ and negative colonoscopy results. The five-year mortality rate was 67% among patients with GC (n = 48), 3.0% among participants in the gastric screening program (n = 1080) and 0.09% among the original population participating in the colorectal screening program (n = 35,640). Intestinal-type adenocarcinoma was the most frequent type (77%), with the tumor most commonly located in the antrum (41%). Overall, 334 participants had PGL, and seven of them (2.1%) showed neoplastic progression to GC during the follow-up. Factors associated with GC were age, Helicobacter pylori (HP) infection, obesity (body mass index BMI > 30), smoking, a diet of salty preserved foods, low income and a family history of GC. (4) Conclusions: The incidence of GC is low in central SA, but screening for PGL and GC among patients with HSgFOBT+ and negative colonoscopy may prevent or result in the early treatment of GC. HP eradication, normal body weight, not smoking and adhering to a healthy diet can reduce the risk of GC. The resulting data provide important input for the improvement of national guidelines.

**Keywords:** gastric cancer screening; incidence; premalignant gastric lesions; gastroscopy; epidemiology; risk factors

## 1. Introduction

Gastric cancer (GC) globally ranks fifth in terms of incidence, corresponding to 5.6% of all cancers in 2020, while it is fourth in terms of cancer-related mortality, corresponding to 7.7% of all cancer deaths [1,2]. It represents 7.1% of all cancers in males and 9.1% in females, and it is the cause of 4% of all cancer deaths in males and 6% in females.

According to the GLOBOCAN 2020 report, GC represents 2.4% of all cancers in Saudi Arabia (SA),is the cause of 4% of all cancer deaths, has a five-year prevalence (all ages) of 3.21 per 100,000 population and a cumulative risk of 0.31% [3];this is lower compared to the worldwide cumulative risk of 1.31% [2–4]. The GLOBOCAN report [2,3] also emphasized the existence of disparity and heterogeneity with respect to the burden of GC indifferent regions and found that the age-standardized incidence rate was higher in eastern Asia, followed by central and eastern Europe and South America. On the other hand, the lowest incidence rates are in southern Africa, western Africa, North America, central and northern Africa, Australia and New Zealand [2,4]. In the Arab world, GC is more frequent in Oman and Yemen (incidence rates of 8 and 7.1 per 100,000, respectively) compared to other countries in the region, and it is less frequent in Comoros (1.3 per 100,000), Sudan (2.5 per 100,000), Kuwait and SA (2.7 per 100,000) [5].

Over the past 50 years, the histopathological classification of GC has predominantly relied on Lauren's criteria for histological classification [6]. One of the most common histological types of GC is adenocarcinoma, responsible for 90–95% of all cases. Adenocarcinomas are typically classified into two subtypes based on Lauren's classification: diffuse and intestinal. Additionally, there is a mixed subtype that exhibits the characteristics of both diffuse and intestinal types [6]. Other less common types of GC are primary gastric lymphoma, gastrointestinal stromal tumors (GIST) and neuroendocrine tumors [6].

GC is frequently asymptomatic in its early stages. In many cases, the presence of nonspecific symptoms does not prompt immediate investigation [7]. According to the guidelines of the European Society of Medical Oncology (ESMO), the gold standard for diagnosis remains upper gastrointestinal (GI) endoscopy, followed by the pathological examination of biopsy specimens obtained during the endoscopic procedure [7]. Histology and molecular interpretation require multiple biopsies [6–8], especially for ulcerated lesions [8,9]. Due to its invasiveness, such screening may be challenging to use on the general population. The mucosal surface of the stomach can be evaluated in detail by using electronic or virtual chromo-endoscopy in combination with magnifying endoscopy. These techniques enhance the detection rate of precancerous gastric lesions and early gastric cancer [10], which may result in better prognoses and improved overall survival rates; asymptomatic patients have a five-year survival rate of above 90% [11]. Endoscopic ultrasonography (EUS) is another valuable tool for identifying infiltrated regions of the gastric wall [12].

Unfortunately, no blood surrogate markers are available for the early diagnosis of GC. Several countries have evaluated the potential usefulness of measuring serum pepsinogen (PG) levels. However, its application in GC screening and detection is not yet widespread, and relying solely on pepsinogen tests is inadequate for GC detection [13]. The serological screening of serum pepsinogen with Helicobacter pylori (HP) whole cell and cytotoxin-associated gene A (CaGA) is a non-invasive test that is used for the prevention of GC; however, exploited data remain controversial [13,14]. The identification of molecular profiles in GC has provided valuable insights into the potential identification of clinically relevant biomarkers [15].

Despite the decline in the incidence and mortality rates of gastric cancer in recent years, it continues to be a significant cause of cancer-related deaths [1]. Approximately 60% of individuals diagnosed with GC cannot receive curative therapy due to the late presenting symptoms of the disease or the presence of other comorbidities [16]. The early detection of gastric cancer allows for better treatment with the use of minimally invasive procedures such as endoscopic mucosal resection (EMR) and endoscopic submucosal resection (ESD) [17]. GC can have various causes, including genetic, racial and environmental factors. Consequently, the prevalence of GC varies between geographical regions and among developed and developing countries. Updated European guidelines [7] recommend extending cancer screening to GC and other targeted cancers.

In areas with high gastric cancer incidence and mortality rates, screening for HP infections and conducting a surveillance of precancerous stomach lesions are recommended.

Endoscopic surveillance is recommended for patients with GC precursor lesions (GCPL), including intestinal metaplasia (IM), as well as for those with a family history of gastric cancer, incomplete-type IM, or persistent H. pylori–associated gastritis [7].

In SA, the incidence rates of cancer have been notably affected by rapid urbanization, lifestyle factors, and an increase in life expectancy [18]. However, there may be regional variations in the incidence of exposure to trigger factors such as water-pipe smoking, a change from a Mediterranean diet to a fast-food diet, and a high incidence of HP infection [18]. Few hospital-based studies have been conducted in SA to examine the patterns and clinical pathology of GC [18], while no data exist on PGL.

Upon recognizing the increasing burden of GC, our objective was to implement a GC screening program in the central regions of SA despite the country being classified as having a low incidence of the disease. Studying the epidemiology of GC in different geographical regions across the country is important for the control of mortality and the implementation of efficient management and treatment strategies. This initiative will contribute to the development of screening recommendations specifically tailored for low-incidence populations, such as those in SA, with a focus on GC. We aimed to assess the incidence of sporadic GC, its clinicopathological profiles, and factors associated with GC in central SA among subjects who tested positive in the high-sensitivity guaiac-based fecal occult blood test (HSgFOBT) and had negative colonoscopy results. Additionally, we aimed to identify and follow up subjects with PGL in order to assess neoplastic progression.

## 2. Subjects and Methods

### 2.1. Study Design

A population-based prospective cohort study of asymptomatic Saudi individuals aged 45–75 years, with registered addresses in Al-Kharj, was conducted from January 2017 to May 2023. Initially, a colorectal cancer screening program, previously carried out in Al-Kharj, Riyadh province in central SA, was adopted as described elsewhere [19]. Individuals with positive HSgFOBT and negative colonoscopy results were invited to participate in the GC screening program involving upper gastrointestinal endoscopy.

Individuals registered voluntarily via Public Health Registries and were recruited randomly based on their birth month [19]. These individuals were invited to participate in the program through their general practitioner, internist or by email. The endoscopy report databases of several medical facilities, including the Endoscopy Unit of King Khaled Hospital, Prince Sultan Center for Health Care, Prince Sattam bin Abdulaziz University Hospital, Prince Sultan Military Medical City of Riyadh and Al Kharj Military Hospital, were used to collect data on the participants.

Sociodemographic information, such as age, sex, educational level, income, family history of cancer as well as area of residence, was collected through supplementary questionnaire. Additionally, data regarding other risk factors for GC, including HP infection, lifestyle factors such as smoking (e.g., tobacco, cigarettes, water pipe smoking or shisha), diet with foods preserved by salting (e.g., salted fish and meat, pickled vegetables, etc.) and obesity were recorded. While alcohol is an established risk factor for GC, SA is a country where alcohol consumption is forbidden; thus, this risk factor was not applicable in our study. The questionnaire is presented in Supplementary Materials.

The five-year outcomes of the gastroscopy screening for PGL associated with GC were recorded according to the updated guidelines of the European Society for Medical Oncology (ESMO) [7]. These outcomes serve as a valuable tool for the prospective enhancement of early gastric cancer diagnosis using endoscopic surveillance. Based on the recommendations for the management of epithelial precancerous conditions and lesions in the stomach (MAPS II) by the European societies [20], patients with previous diagnoses of atrophic gastritis or intestinal metaplasia (IM) and a family history of gastric cancer, incomplete IM or persistent H. pylori-associated gastritis or dysplasia were offered endoscopic surveillance. These surveillance sessions involved guided biopsies and were conducted after three years to monitor the neoplastic progression. The time interval between surveillance

endoscopies was 1 year for cases with low-grade dysplasia (LGD) and 6 months for those with high-grade dysplasia (HGD). When a visible lesion was identified, the patient was obliged to undergo endoscopic resection as soon as possible.

### 2.2. Ethical Considerations

The present study adhered to the STROBE statement. The study design, protocols and informed consent procedures were approved by the Ethics Committee of the PSAU University of Medical Sciences (approval code: PSAU/COM/RC/IRB/p/67).

### 2.3. Study Participants

The study involved asymptomatic Saudi individuals aged 45–75 years; these individuals were eligible for colorectal cancer screening at the age of 45 years and had tested positive for HSgFOBT but had no colonoscopy findings. The inclusion criteria were as presented in our previous manuscript about the colorectal cancer screening program [19], while in addition to previously used exclusion criteria patients with a previous history of GC, known PGL, portal hypertension, dyspepsia, reflux, anemia, weight loss, upper bleeding gastrointestinal symptoms, dysphagia or any other symptom concerning the upper GI tract were also excluded. All participants underwent upper gastrointestinal endoscopy. Those identified with PGL were offered endoscopic surveillance every 3 years according to international guidelines [7,20].

### 2.4. Screening Methods

2.4.1. Invasive Screening

The endoscopic procedure was performed using an endoscope with recent advanced image zooming, high definition and virtual or electronic chromoendoscopy for the detection, characterization and treatment of PGL and early GC. In this study, we used the GIF-H190 models of Olympus Co., Tokyo, Japan. The endoscopic procedure was performed by a single experienced endoscopist. In cases where the upper GI tract endoscopy could not be completed successfully, the procedure was repeated, with the participant under sedation using propofol.

2.4.2. Tissue Sampling

Tissue sampling was performed selectively when abnormal mucosal findings were observed during the endoscopy. For PGL evaluation, five biopsies were taken from at least two topographic sites (antrum and incisura angle as well as corpus—lesser and greater curvature) and deposited in two separate, labeled vials [20]. This allowed for both diagnostic confirmation and risk stratification of the progression to cancer. The collected tissue samples were then fixed in 10% neutral buffered formalin for preservation. Each tissue sample was appropriately labeled with the subject's identifier and the date of collection. To minimize measurement bias, one pathologist who was blinded to the study purpose and protocol analyzed the tissue samples and also performed the re-evaluations of each baseline histology group of PGL. The re-evaluated tissue samples showed three outcomes: (a) more progression in advanced lesions in re-endoscopies; (b) regression, i.e., less advanced lesions; or (c) no change, i.e., neither regression nor progression (stable) in the lesions. Four potential outcomes were considered based on the histological diagnosis of GC: (1) adenocarcinoma, (2) gastrointestinal stromal tumor (GIST), (3) gastric neuroendocrine tumor and (4) malignant lymphoma. In the case of PGL, possible outcomes based on mucosal histology findings may include chronic atrophic gastritis with or without intestinal metaplasia (IM), as well as varying degrees of dysplasia, such as LGD or HGD.

2.4.3. EBV and HP Detection

The Epstein–Barr virus (EBV) was detected in tumor tissues via in situ hybridization for small RNAs (EBERs) and immunohistochemistry for EBV latent protein (LMP1 and

LMPA2), as described elsewhere [21]. HP was detected based on hematoxylin–eosin (H–E) staining [22].

### 2.5. Statistical Analysis

Statistical analyses were performed using SPSS version 24 (Chicago, Armonk, NY, USA). A *p*-value < 0.05 was used to determine statistical significance. Continuous variables are expressed as means and standard deviation (SD). Categorical variables are presented as numbers (n) and their respective frequencies as percentages (%). To compare between groups chi-square test was used. Logistic regression analysis was performed to identify factors associated with GC diagnosis. Sociodemographics, body mass index (BMI), smoking status, diet, family history of GC, HP infection, EBV infection and histology of gastric mucosa were used as independent variables in the logistic regression analysis.

### 3. Results

Figure 1 presents the flowchart for the study participation. Out of the 35,640 participants in the colorectal cancer screening program, 1242 were eligible to participate in the GC screening program because of HSgFOBT+ and negative colonoscopy results. Of those, 1080 (87%) were screened for GC (participation rate: 87%). The baseline characteristics of the study participants are shown in Table 1. Their average age was 59 (SD11.3) years, and570 (52%) were male. Approximately half were obese, and more than one-third were current smokers.

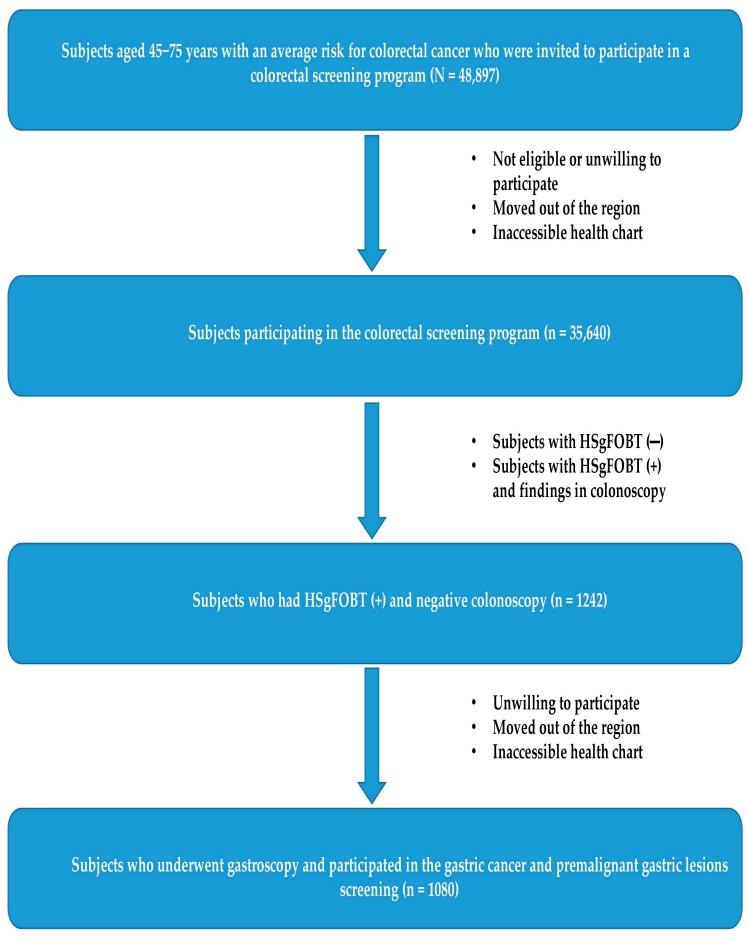

**Figure 1.** Participants flowchart in the prospective cohort study, 2017–2022.

**Table 1.** Baseline characteristics of study participants (N = 1080).

| Factor | N | % |
|---|---|---|
| Age(mean, SD) | 59 ± 11.3 | |
| Age groups | | |
|     45–55 years | 361 | 33.4 |
|     56–65 years | 351 | 32.5 |
|     66–75 years | 370 | 34.3 |
| Gender | | |
|     Female | 510 | 47.2 |
|     Male | 570 | 52.8 |
| Education | | |
|     University College | 233 | 21.6 |
|     Secondary education completed | 331 | 30.6 |
|     Primary education completed | 220 | 20.4 |
|     Primary education not completed | 296 | 27.4 |
| Monthly income | | |
|     >20,000 SAR | 216 | 20.0 |
|     10,000–20,000 SAR | 382 | 35.4 |
|     <10,000 SAR | 482 | 44.6 |
| BMI | | |
|     Normal (18.5–25 kg/m$^2$) | 499 | 46.2 |
|     Overweight (25–30 kg/m$^2$) | 104 | 9.6 |
|     Obese (>30 kg/m$^2$) | 477 | 44.2 |
| Smoking status | | |
|     Never smoked | 554 | 51.3 |
|     Former smoker | 148 | 13.7 |
|     Current smoker | 378 | 35.0 |
| Diet | | |
|     Fresh fruits, vegetables, unprocessed wheat products | 359 | 33.2 |
|     Animal products, hot spices, canned and fermented foods | 721 | 66.8 |
|     Nutritional salty preserved products | 673 | 62.3 |
| Family history of GC | | |
|     No | 1048 | 97.0 |
|     Yes | 32 | 3.0 |

The delay from the time of obtaining a negative colonoscopy result to the performance of gastroscopy varied from zero (performed within the same day, after the colonoscopy) to a maximum of eight weeks after the invitation to participate in the program. During the study period, no complications in the gastroscopy procedures were reported, and there were no instances of further delays despite the challenges posed by the COVID-19 pandemic. However, it should be noted that the participation rate during the pandemic period of 2019–2020 was 30% lower compared to the pre-pandemic period. Of all participants, 3% (33/1080) underwent a second scheduled gastroscopy under propofol sedation because of improper preparation and the presence of food in the stomach.

Table 2 shows HP infection and PGL among the study participants. PGL was detected in 334 subjects (30.9%). Specifically, atrophic gastritis was diagnosed in 93 participants (27.8% of PGL patients), intestinal metaplasia in 173 (51.8%), LGD in 65 (19.5%) and HGD in 3 (0.9%). Histology showed HP infection in approximately 60% of all participants (with 74% of non-cardia gastric cancer cases showing HP infection) and in 67.7% of those with PGL (n = 226). Seven subjects with PGL developed GC (2.1%) during the follow-up period. The overall risk for neoplastic progression was 0.4% per year. No change was observed in the histology performed on180 subjects (54%). After successful HP eradication, regression was observed in 55 subjects with LGD, in 72 subjects with intestinal metaplasia and in 20 subjects with atrophic gastritis (in total, 44% of patients with PGL).

**Table 2.** Helicobacter pylori and Epstein–Barr virus infection, as well as premalignant gastric lesions detected in histology of study participants.

|  | N | % |
|---|---|---|
| HP infection |  |  |
| No | 431 | 39.9 |
| Yes | 649 | 60.1 |
| EBV infection |  |  |
| No | 44 | 48 |
| Yes | 4 | 48 |
| Premalignant gastric lesions |  |  |
| Normal mucosa | 397 | 36.8 |
| Chronic gastritis | 683 | 63.2 |
| Chronic atrophic gastritis | 93 | 8.6 |
| Intestinal metaplasia | 173 | 16.0 |
| Dysplasia | 68 | 6.3 |

GC was diagnosed in 48 participants, with a 5-year prevalence rate of 4.4% (48/1080). The incidence rate of GC in the original population participating in the colorectal screening program (N = 35,640) was 26.9 new cases per 100,000 population per year, while it was 7.4 per 1000 persons per year for the 1080 subjects with HSgFOBT+ and negative colonoscopy. The 5-years prevalence rate was 134.7 cases per 100,000 population. Seven of the detected GC cases were subjects with PGL during the follow-up screening, with a risk of neoplastic progression of 0.4% per year. Over the 5-year period, 32 subjects died from GC; a 5-year mortality rate of 67% was determined for patients with GC (n = 48), 3.0% for participants in the gastric screening program (n = 1080) and 0.09% for the original population participating in the colorectal screening program (N = 35,640). None of the patients with PGL who developed GC died. The mean (SD) age of the subjects diagnosed with GC was 61 (15.6) years, with male predominance; out of the 48 participants with GC, 34 were male (6% of the total male participants, n=570) and 14 were female (2.8% of the total female participants, n=510) (*p* < 0.05). The age distribution of GC cases is shown in Figure 2, with the highest prevalence in the age group of 71–75 years (*p* = 0.001). As shown in Figure 3, the most common histological type of GC is primary adenocarcinoma at 85% (41/48), followed by lymphomas at 10% (5/48), gastrointestinal stromal tumors at 2.5% (1/48) and gastric neuroendocrine tumors at 2.5% (1/48). According to Lauren's classification of adenocarcinomas, intestinal adenocarcinoma is the most frequent (77%), followed by diffuse adenocarcinoma (12%) and indeterminate (6%), as shown in Figure 4.

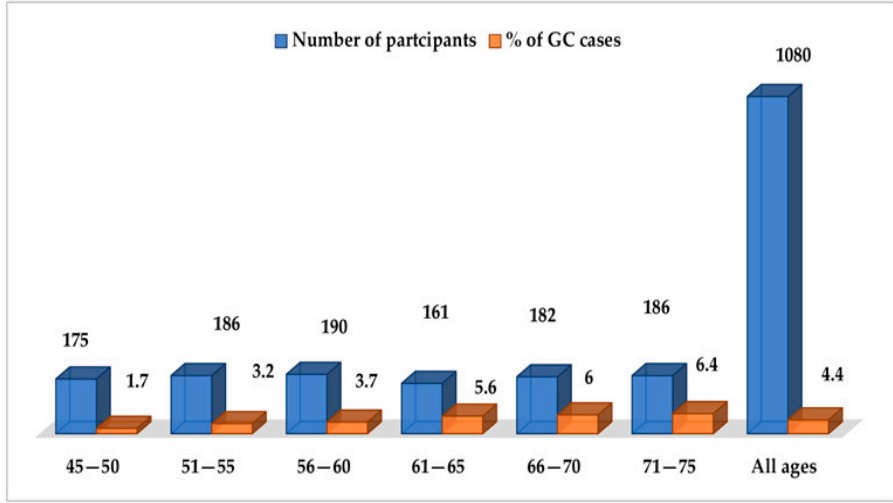

**Figure 2.** Age distribution of sporadic gastric cancer.

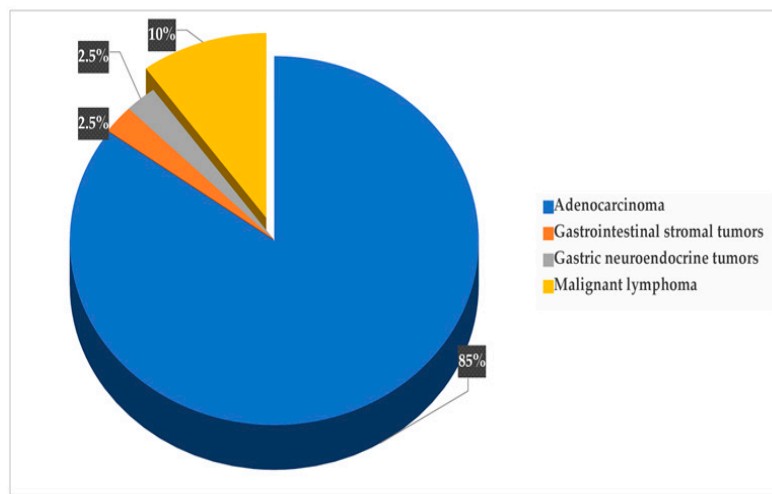

**Figure 3.** Distribution of cancer types.

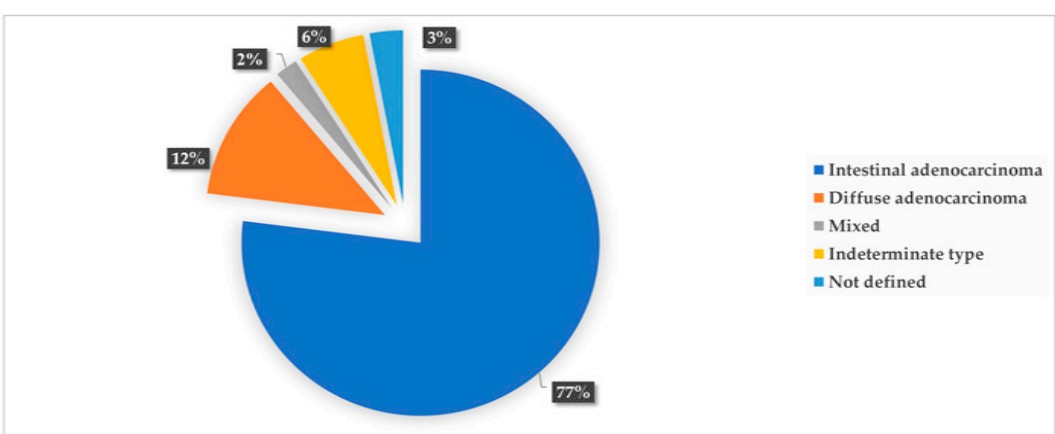

**Figure 4.** Frequency of Lauren's histopathological types of gastric cancer.

As presentedin Figure 5, GC tumors were most commonly located in the gastric antrum–pylorus (non-cardia) at 41% (20/48), followed by the corpus at 23% (11/48), the cardia at 19% (9/48) and the fundus at 17% (8/48).

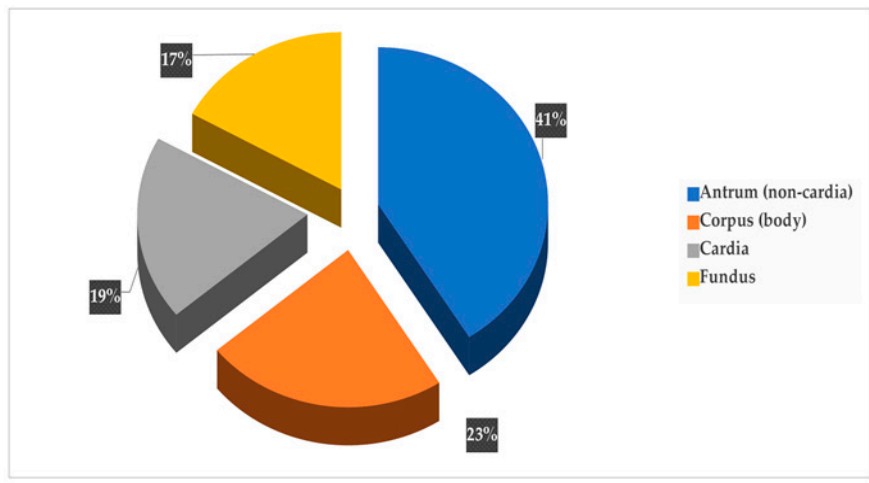

**Figure 5.** Site distribution of gastric cancer.

The factors associated with GC are shown in Table 3. Higher age, higher income, lower education, obesity, smoking, diet with salty preserved products, having mucosal

dysplasia, positive family history of GC, HP infection and EBV positivity in the tumor were significantly associated with GC.

**Table 3.** Factors associated with gastric cancer diagnosis, examined by binary logistic regression analysis.

| Factor | OR | 95% CI | *p* |
|---|---|---|---|
| Gender | | | |
| Female | Ref. | | |
| Male | 2.28 | 1.67–2.4 | 0.003 |
| Age | | | |
| 45–55 | Ref. | | |
| 56–65 | 1.08 | 1.10–2.08 | <0.001 |
| 66–75 | 1.23 | 1.12–2.63 | 0.004 |
| BMI | | | |
| Normal (<25 kg/m$^2$) | Ref. | | |
| Overweight (25–20 kg/m$^2$) | 0.8 | 0.93–4,1 | 0.58 |
| Obesity (>30 kg/m$^2$) | 2.82 | 1.43–5.59 | 0.001 |
| Education | | | |
| University College | Ref. | | |
| Secondary education completed | 0.73 | 0.35–1.51 | 0.610 |
| Primary education completed | 1.89 | 1.00–3.57 | 0.002 |
| No complete primary education | 2.11 | 2.64–9.98 | <0.001 |
| Monthly income | | | |
| >20,000 SARs | Ref. | | |
| 10,000–20,000 SARs | 2.33 | 1.24–4.39 | 0.007 |
| <10,000 SARs | 3.85 | 1.87–7.92 | 0.006 |
| Smoking status | | | |
| Non smoker | Ref. | | |
| Former smoker | 0.79 | 0.28–2.24 | 0.179 |
| Current smoker | 4.00 | 2.05–7.81 | 0.002 |
| Diet | | | |
| Fresh fruits, vegetables, unprocessed wheat products | Ref. | | |
| Animal products, hot spices, canned and fermented foods | 2.90 | 1.21–6.97 | <0.001 |
| Nutritional salty preserved products | 1.87 | 1.01–3.86 | 0.003 |
| Histology | | | |
| Normal mucosa | Ref. | | |
| Chronic atrophic gastritis | 1.25 | 0.24–6.48 | 0.060 |
| Intestinal metaplasia | 2.54 | 0.49–13.27 | 0.070 |
| Dysplasia | 6.78 | 1.49–30.94 | 0.010 |
| Family history with GC | | | |
| No | Ref. | | |
| Yes | 4.55 | 1.67–12.36 | <0.001 |
| Helicobacter pylori infection | | | |
| No | Ref. | | |
| Yes | 8.39 | 1.10–3.30 | 0.001 |
| EBV | | | |
| No | Ref. | | |
| Yes | 2.17 | 1.19–2.87 | 0.001 |

Abbreviations: OR (odds ratio), 95% CI (95% confidential interval).

## 4. Discussion

GC incidence varies significantly worldwide. Evidence suggests that there is no benefit in the widespread implementation of upper gastrointestinal (GI) endoscopy screening for GC in the general population of geographic regions with low incidence rates. The potential benefits of such screening in countries with intermediate risks are also uncertain [6,20,23,24]. However, a recent cost–utility analysis that employed a combination of endoscopy screening with gastroscopy and colonoscopy in volunteers aged between 50 and 75 years suggested that this combined screening may be cost-effective [25]. The current understanding of the role of upper GI endoscopy in preventing GC is that it serves as a valuable screening tool for predefined high-risk individuals [6,23]. The updated EU recommendations recently included GC screening, given the growing awareness of the GC burden [6]. Similar to the guidelines from multidisciplinary European societies, the British Society of Gastroenterology (BSG) guidelines also recommend upper GI endoscopy screening for predefined high-risk individuals aged over 50 years. These high-risk factors

include being male, smoking, having pernicious anemia and/or having a family history of gastric cancer, along with being subjected to a follow-up for PGL [7,23]. In low-incidence regions, there are fewer data available on the endoscopic assessment and surveillance of PGL, which may negatively affect the yield of surveillance. Accordingly, after 2017,we implemented a GC screening program in Al-Kharj, Riyadh province, a central rural area of SA, for a period of five years in order to assess the prevalence of GC in this special target population, its clinicopathological profiles, as well as PGL and its association with risk factors. To the best of our knowledge, this is the first study to identify participants with PGL and to help in the prevention of GC by conducting surveillance examinations of these individuals every 3 years, according to international guidelines.

The most prevalent histological type of GC observed in our study was primary adenocarcinoma, which is consistent with the findings of a recent meta-analysis of GC in the Arab world [5] and a study conducted in northern Jordan [26]. Additionally, other histological types observed among GC patients in our study included malignant stromal tumors, lymphomas and carcinoids, which aligns with the results of a study conducted in Jordan [27]. Among the adenocarcinoma subtypes, the intestinal type, according to Lauren's classification, was the most common [28,29], followed by the diffuse type, mixed type and adenosquamous carcinoma [30,31]. These findings are consistent with the patterns observed in GC classification studies conducted in Arab countries [5].

In our study, we assessed tumor localization, and our findings are consistent with the majority of studies conducted in the Arab world as well as a recent meta-analysis of GC in the region [5]. The most frequent tumor location observed was the third distal region, which includes the antrum and pylorus [27,30], followed by the body (middle part) and proximal sites [27]. The two topographical sub-sites of gastric cancer, the cardia (distal stomach) and non-cardia (proximal stomach), exhibit distinct characteristics that are correlated with risk factors, carcinogenesis, and epidemiology across geographic regions. Chronic HP infection is widely recognized as the primary etiology of non-cardia GC [32,33]. Our study observed similar results, with 74% of non-cardia GC cases showing HP infection. This finding is consistent with the global trend that non-cardia GC, which is more prevalent in East Asia and Latin America, accounts for approximately 80% of gastric tumors worldwide [34]. Overall, the prevalence in the cohort of chronic HP infection was 60%. The prevalence of HP infection varies in different studies conducted in Saudi Arabia, ranging from 25% to 82.2% [35]. These variations could be attributed to the differences in the study population in terms of age groups and associated functional dyspepsia. A 2022 study conducted in Taif reported that almost two-thirds (65.6%) of patients with HP infection had active chronic gastritis, which is consistent with the results of our study [36].

In this study, other risk factors, such as obesity (BMI > 30), smoking status, nutrition/consuming salty preserved products, low socioeconomic status, having a family member with GC and the age groups of 56–65 and 66–75 were positively associated with GC. Our findings with regard to smoking being a significant risk factor for gastric cancer (GC) align with the results of several Arab studies [34], which increases the odds of developing GC by a factor of three [37]. Similar observations have been made in Lebanon, where smoking has also been recognized as a major risk factor for GC [38].

Non-cardia GC, which is the cause of almost 80% of gastric tumors worldwide, is more prevalent in East Asia and Latin America and has been linked to other risk factors such as alcohol consumption, high salt intake and low consumption of fibers such as fruits and vegetables. In contrast, proximal (cardia) gastric cancer is primarily correlated with obesity and gastroesophageal reflux disease (GERD). It is more commonly found in North America and Western Europe [31]. Alcohol is not consumed in SA and is therefore not included in our study as a risk factor. High salt consumption and obesity were positively associated with GC, similar to the findings of studies carried out in North America, Europe, East Asia and Latin America. EBV as a risk factor for gastric GC is more frequently present in GC of the fundus or body (62%); its prevalence seems to be similar to thatin Asia, Europe and the Americas at about 8.7% [39,40]. Two Tunisian studies found the frequency of EBV

associated with GC to be 10% and 14.8% [41,42]. In our study, EBV associated with GC had a lower prevalence (8.5%) and mostly present in the proximal stomach (57%).

The Correa cascade is a well-known model that describes the stepwise progression of normal mucosa through chronic gastritis (chronic inflammation of the gastric mucosa), to PGL such as mucosal atrophy (loss of gastric glands) and intestinal metaplasia (substitution of gastric epithelium by intestinal epithelium), to dysplasia (intraepithelial neoplasia) and ultimately, to carcinoma in a multistep process. Little is known about PGL and how such lesions can further progress to dysplasia and eventually develop into GC. In SA, no data were available. Following European and British guidelines [7], we attempted to shed some light on the identification and surveillance of premalignant lesions, such as atrophic gastritis, incomplete intestinal metaplasia and mild to moderate dysplasia, dysplasia frequencies and the progression of GC. In our study, almost one-third of our subjects were identified to have PGL. Neoplastic progression was 0.4% per year, as reported in other studies. Most of the participants with PGL showed stable disease or histological regression which is likely caused by HP eradication but could also be false negatives during histologic sampling. In China, studies showed atrophic gastritis to have a prevalence of 25.8%, with 17.7% endoscopically diagnosed, resulting in the prevalence of IM beingat 23.6% and dysplasia at 7.3% among patients with PGL [15,43]. In the Netherlands and Norway, which are low-GC-incidence countries, neoplastic progression was 0.3% per year, atrophic gastritis was at 4%, IM prevalence was at87% and dysplasia was at 9%, while 26% of patients with PGL had HP infection [44]. In Sweden, which comprises a low-risk western population, the annual crude incidence of gastric cancer for those with normal mucosa was 20 per 10,000 population per year, 59 for chronic gastritis, 100 per 263 for atrophic gastritis, 129 for intestinal metaplasia and 263 for those with dysplasia [45].

One of the limitations of our study is the small number of cases and the limited follow-up period. A worthwhile challenge would be to examine neoplastic progression and its association with risk factors for a longer period. More cases and a longer follow-up period would provide input for the national guidelines concerning the early management of these subjects. The study was also confined to a region in central SA, potentially limiting the generalizability of the findings to the whole country. Such a screening program should be expanded throughout the country. Ethnicity was not examined as a confounder; however, the vast majority of Saudis are Arabs (approximately 90%), with a small proportion being Afro-Arabs [46]. Genetic risk factors were not included in this study. GC shows familial aggregation in ~10% of cases, and there is an inherited genetic predisposition with known specific gene mutations in up to 3% of cases [47]. The role of low-penetrance genes in GC is an important step in genetic counselling since an early therapeutic endoscopic intervention could be offered. Therefore, future studies should include genetic factors as confounders. Another limitation is that the tissue samples were not taken from the same mucosal region of the stomach during there-endoscopies. PGLs are usually patchy lesions, and there is always a risk of obtaining false negative results when a re-endoscopy is performed. Additionally, recent studies have highlighted a concerning increase in non-cardia GC cases among young individuals, particularly those below the age of 50, in countries such as the UK and the US, which typically have a low prevalence of HP infection [48]. In our epidemiological study, we included subjects aged $\geq 45$ years based on established guidelines. While we find this adequate, particularly for a low-incidence country such as SA, future studies should examine the value of screening programs for younger age groups. Currently, there is no reason to screen the general population for GC in low-incidence countries according to the guidelines [7], although few data are available on the endoscopic assessment and surveillance of PGL. Precancerous lesions can provide crucial insights into GC development and progression. The absence of such data may affect the understanding of the disease pathogenesis [49].

## 5. Conclusions

The current prospective five-year study summarized the characteristics of GC tumors in central Saudi Arabia, a low-incidence country for GC. The new strategy showed a relatively higher incidence of GC for the high-risk population that was selected for screening. The early diagnosis of GC has an excellent prognosis due to early and effective treatment. Thus, the early diagnosis of GC and the identification and surveillance of PGL have potential implications for clinical practice and public health in terms of GC prevention and mortality reduction in low-incidence areas. Upper endoscopy at the setting of colonoscopy may be an alternative strategy for selected patient populations with high risk factors in low-incidence regions. The strategy of screening subjects with HSgFOBT+ and negative colonoscopy allowed for the detection of a higher-risk group that could contribute to a cost-effective approach in the early detection and prevention strategies for GC. Furthermore, risk factors that were identified herein could further enhance prevention strategies but also help isolate higher-risk population subgroups for which screening programs could be developed and further enhanced. The current data may be of considerable use in the implemented and updated national guidelines for GC if direct GC screening is applied in patients aged > 45 years, or even in younger populations with genetic or environmental risk factors and PGL. These data highlight the need for the identification and surveillance of subjects with an increased risk of GC even in low-incidence regions. To achieve a more comprehensive and accurate understanding of gastric carcinogenesis, scientists must conduct further prospective studies and collect more national data.

**Supplementary Materials:** The following supporting information can be downloaded at: https://www.mdpi.com/article/10.3390/curroncol30090605/s1, File S1: questionnaire.

**Author Contributions:** Conceptualization, G.Z., A.A. and O.A.; methodology, G.Z. and A.A.; software, J.A. and S.A.-G.; validation, G.Z.; formal analysis, J.A., S.A.-G. and A.A.; investigation, G.Z., A.A. and O.A.; resources, J.A., S.A.-G. and G.Z.; data curation, G.Z., J.A. and S.A.-G.; writing—original draft preparation, A.A. and G.Z.; writing—review and editing, A.A. and G.Z.; visualization, J.A. and S.A.-G.; supervision, A.A. and G.Z.; project administration, J.A. All authors have read and agreed to the published version of the manuscript.

**Funding:** This research was funded by the Prince Sattam bin Abdulaziz University, specifically from the Deanship of Scientific Research and College of Medicine in the Kingdom of Saudi Arabia, grant number [2022/03/19471].

**Institutional Review Board Statement:** The study was conducted in accordance with the Declaration of Helsinki and approved by the Institutional Review Board of Prince Sattam bin Abdulaziz University (PSAU/COM/RC/IRB/p/67, 30 March 2016).

**Data Availability Statement:** Data are available upon reasonable request.

**Acknowledgments:** We would like to thank Pavlos Nikolaidis for the statistical analysis, Konstantinos Farsalinos for the report analysis, Daadour Moataz, Albadrani Ahmed, Aldosarry Khaled, AlShehri Abdullah, Bawazir Abdullah, Alfuhaid Mohammed, Alonezi Khalid, Alotabi Badrand and Alhammad Abdullah for their major contribution in data collection and analysis.

**Conflicts of Interest:** The authors declare that they have no affiliations with or involvement in any organization or entity with any financial interest in the subject matter or materials discussed in this manuscript.

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
