# Peer review of "Epidemiology, Risk Factors for Gastric Cancer and Surveillance of Premalignant Gastric Lesions: A Prospective Cohort Study of Central Saudi Arabia"

_curroncol, doi:10.3390/curroncol30090605_

Round 1

Reviewer 1 Report

The manuscript under review provides a noteworthy contribution to the field with its original investigation into gastric cancer (GC) in central Saudi Arabia - an area currently underrepresented in research. This five-year retrospective cohort study adds valuable knowledge to understanding GC in the region, making the manuscript an original and novel endeavor.

However, while the study's design and regional focus bring an element of novelty, several areas could be improved upon. Below are the concerns, both major and minor:

  1. The retrospective cohort design is generally suitable for investigating disease progression and risk factors. However, the study's sample lacks diversity, particularly regarding the age range of participants. Considering the emerging trend of GC incidence in younger populations, expanding the age range would enhance the study's external validity.
  2. The omission of genetic risk factors is a notable gap. Given that familial clustering and specific gene mutations are known to impact GC risk, neglecting this aspect may have resulted in a less comprehensive understanding of GC. In addition, ethnic background and demographic stratification were not adequately considered in the analysis. 
  3. The findings' generalizability may be limited due to the study's geographical confinement to central Saudi Arabia. Expanding the study to a broader geographical scope could provide a more nationally representative picture of GC.
  4. Precancerous lesions can provide crucial insights into GC development and progression. The absence of this data affected the understanding of the disease pathogenesis. 
  5. While the language is generally understandable, there are a few instances where the wording could be clearer. For example, the explanation of the Correa cascade could benefit from more accessible language to make the process more understandable to readers unfamiliar with this model. Note that this is one of many instances where linguistic clarity is required. 
  6. Even though alcohol consumption isn't typical in Saudi Arabia, it could be beneficial to include it compared with other studies in regions where alcohol consumption is a recognized risk factor.
  7. The screening restriction for those who underwent a colonoscopy after a positive FOBT might have led to underestimation. A broader screening protocol could better reflect the actual prevalence of gastric cancer and its precursor lesions.
  8. While the manuscript's originality is commendable, it would benefit from a more thorough discussion of its limitations. An in-depth examination of the study's constraints will provide a more balanced view and guide future research in addressing these issues. The constraints tied to the study's geographical scope, the age range of participants, and the absence of genetic risk factors should be explicitly discussed.

The manuscript provides valuable data on GC in central Saudi Arabia. However, addressing the above concerns could significantly enhance the study's reliability, robustness, and applicability to the broader population.

The manuscript's language is generally clear and understandable, facilitating comprehension of the study's technical aspects. However, there are areas where the language could be further refined and simplified to improve accessibility and readability. Here are some specific suggestions:

  1. There are instances where more precise language could aid understanding. For instance, in discussing the Correa cascade, the language used to describe the progression from non-atrophic gastritis or atrophic gastritis to gastric cancer could be more specific. 
  2. Ensure that there is logical flow and continuity in your discussion. Some sections abruptly transition to new topics without clear linkages, which can disorient readers.
  3. Some sentences seem overly complex and wordy.
  4. Ensure that all sentences are grammatically correct and that syntax is consistent. Although the manuscript's language is generally good, occasional minor grammatical errors and awkward sentence patterns are there, which can detract from the overall quality of the paper.

Overall, while the language used in the manuscript is understandable, improvements can be made in terms of clarity, coherence, conciseness, and the appropriate use of technical terms. These modifications would significantly enhance the manuscript's readability and accessibility.

Reviewer 2 Report

The article titled "Epidemiology, Surveillance of Premalignant Gastric Lesions, and Risk Factors for Gastric Cancer: A Multicenter Prospective Cohort Study from Low Incidence Central Saudi Arabia." I found your study to be informative and well-conducted, and I would like to commend you on your valuable contributions to the field. I have a few comments and questions regarding your research, which I hope you can address:

1. Methodology:

   a. Could you provide more details about the selection criteria for participants in your study? How did you ensure a representative sample of the population?

   b. In the surveillance of premalignant gastric lesions, what specific techniques or diagnostic tools were used for their detection? Did you employ endoscopy, histopathological analysis, or other modalities?

   c. How frequently were participants followed up for surveillance of gastric lesions? Did the duration of follow-up vary among individuals, and if so, what factors influenced the frequency of surveillance?

2. Results:

   a. Can you elaborate on the incidence of premalignant gastric lesions and gastric cancer in the low incidence area of Central Saudi Arabia? How do these findings compare to regions with higher incidence rates?

   b. What were the most common types of premalignant gastric lesions identified in your study? Did you observe any patterns or associations between specific lesions and subsequent development of gastric cancer?

   c. Did you identify any significant risk factors for the development of gastric cancer in your cohort? Were there any unexpected or novel risk factors that emerged from your analysis?

3. Implications and Future Directions:

   a. Based on your study findings, what are the potential implications for clinical practice and public health interventions in low incidence areas? How can these results contribute to early detection and prevention strategies for gastric cancer?

   b. Were there any limitations or challenges encountered during the study that may have influenced the interpretation of results? How could these limitations be addressed in future research?

   c. Considering the low incidence of gastric cancer in Central Saudi Arabia, do you think there is a need for further investigations to understand the underlying factors contributing to this phenomenon?

it is readable

Round 2

Reviewer 1 Report

I appreciate the authors for their diligent work in revising the manuscript. Despite significant improvements made, there are several issues that this reviewer would like to bring to attention.

  1. Even though an English language edit has been performed, there are still quite a few spelling and grammatical inaccuracies in the manuscript that need to be addressed.
  2. The authors indicated that Table 1 provides data on the ethnicity of the study participants and that the study includes only Saudi nationals. However, being Saudi refers to nationality, not race or ethnicity. Saudi Arabia, like many countries, is ethnically diverse, and it would be useful to have information on the participants' ethnic backgrounds. This would provide a more nuanced understanding of the study's population, particularly as genetic factors and ethnicity can influence the risk and progression of gastric cancer.
  3. Concerning the study's limitations, the authors have addressed some of the issues raised, such as the age range of participants, the geographical confinement of the study, and the omission of genetic risk factors. However, the authors have yet to explicitly discuss the limitations inherent to retrospective studies, including the role of bias and confounding factors.
  4. The quality of the figures, in terms of their visual appearance and data representation, could be improved. This reviewer highly recommends enhancing the figures for better clarity and impact, ensuring they clearly represent the data and relevant statistical analyses.

Although the authors have completed a language edit, the manuscript still contains numerous spelling and grammatical inaccuracies, including in parts that are indicated as revised.
